# Gapless superconductivity in Nb thin films probed by terahertz spectroscopy

Ji Eun Lee [1,5], Joonyoung Choi [2,5], Taek Sun Jung [1], Jong Hyuk Kim[1], Young Jai Choi [1], Kyung Ik Sim [3,4] ✉, Younjung Jo[2] ✉ & Jae Hoon Kim [1] ✉

Time reversal symmetry (TRS) breaking often generates exotic quantum phases in condensed matter. In superconductors, TRS breaking by an external magnetic field not only suppresses superconductivity but also leads to a novel quantum state called the gapless superconducting state. Here we show that magneto-terahertz spectroscopy provides us with a rare opportunity to access and explore the gapless superconducting state of Nb thin films. We present the complete functional form of the superconducting order parameter for an arbitrary magnetic field, for which a fully self-consistent theory is, surprisingly, yet unavailable. We observe a Lifshitz topological phase transition with a vanishing quasiparticle gap everywhere on the Fermi surface, whereas the superconducting order parameter smoothly crosses over from the gapped to the gapless regime. Our observation of the magnetic pair-breaking effects in Nb challenges traditional perturbative theories and opens a pathway to further exploring and manipulating the exotic state of gapless superconductivity.

Condensed matter systems often undergo stunning metamorphoses in their electronic structures when a strong external magnetic field breaks their inherent time-reversal symmetry (TRS). Well-known examples include the magnetic breakdown of superconductivity[1], the quantum Hall effect[2], and magnetically induced novel topological phases such as axionic insulators[3] out of topological insulators and Weyl semimetals[4] out of Dirac semimetals. In the case of superconductors, an external magnetic field above a critical field completely suppresses superconductivity, but, even below the critical field, its pair-breaking effect leads to an exotic form of superconducting state without an energy gap at every point on the Fermi surface[5–7]. This gapless superconducting state still retains the hallmarks of superconductivity, such as zero d.c. resistivity and the Meissner effect, which shows that the true essence of superconductivity lies in the coherent pair correlations rather than in the formation of a gap in the energy spectrum[8,9]. The absence of an energy gap is manifested in several anomalous effects, such as the linear temperature dependence of the heat capacity and the finite density of states even up to zero energy[9–11]. According to recent research, the phase transition from the gapped to the gapless superconducting state in BCS superconductors

belongs to a topological quantum phase transition of Lifshitz type (i.e., of the $2\frac{1}{2}$ order)[12].

In this connection, in contrast to the Bardeen–Cooper–Schrieffer (BCS) theory[13], where a single function $\Delta$ serves both as the order parameter and as the energy gap, the magnetically pair-broken superconducting state is properly characterized by two distinct parameters: $\Delta$ for the order parameter representing the pair correlation and $\Omega_G$ for the energy gap for quasiparticle excitation (hence the name the spectroscopic gap)[8,9]. Pair-breaking sources that separate $\Omega_G$ from $\Delta$ include not only TRS breaking agents, such as an external magnetic field[14–16], paramagnetic impurities[8,9,17–19], and an external d.c. supercurrent bias[20,21] but also ultrastrong terahertz (THz) pulses (inversion symmetry breaking)[22] and normal metals in close proximity[23–25] (Fig. 1a). All these pair-breaking effects can be described in a unified manner in terms of a single pair-breaking parameter $\Gamma$, which controls the order parameter $\Delta$, the spectroscopic gap $\Omega_G$, the density of states $N$, and all thermodynamic functions derived thereof. While $\Gamma$ is proportional to the paramagnetic impurity concentration in the original pair-breaking theory of Abrikosov-Gorkov (AG)[8], for a superconducting thin film in the dirty limit under an in-plane magnetic field

[1]Department of Physics, Yonsei University, Seoul, Republic of Korea. [2]Department of Physics, Kyungpook National University, Daegu, Republic of Korea. [3]Center for Integrated Nanostructure Physics, Institute for Basic Science, Suwon, Republic of Korea. [4]Sungkyunkwan University, Suwon, Republic of Korea. [5]These authors contributed equally: Ji Eun Lee, Joonyoung Choi. ✉e-mail: simki323@gmail.com; jophy@knu.ac.kr; super@yonsei.ac.kr

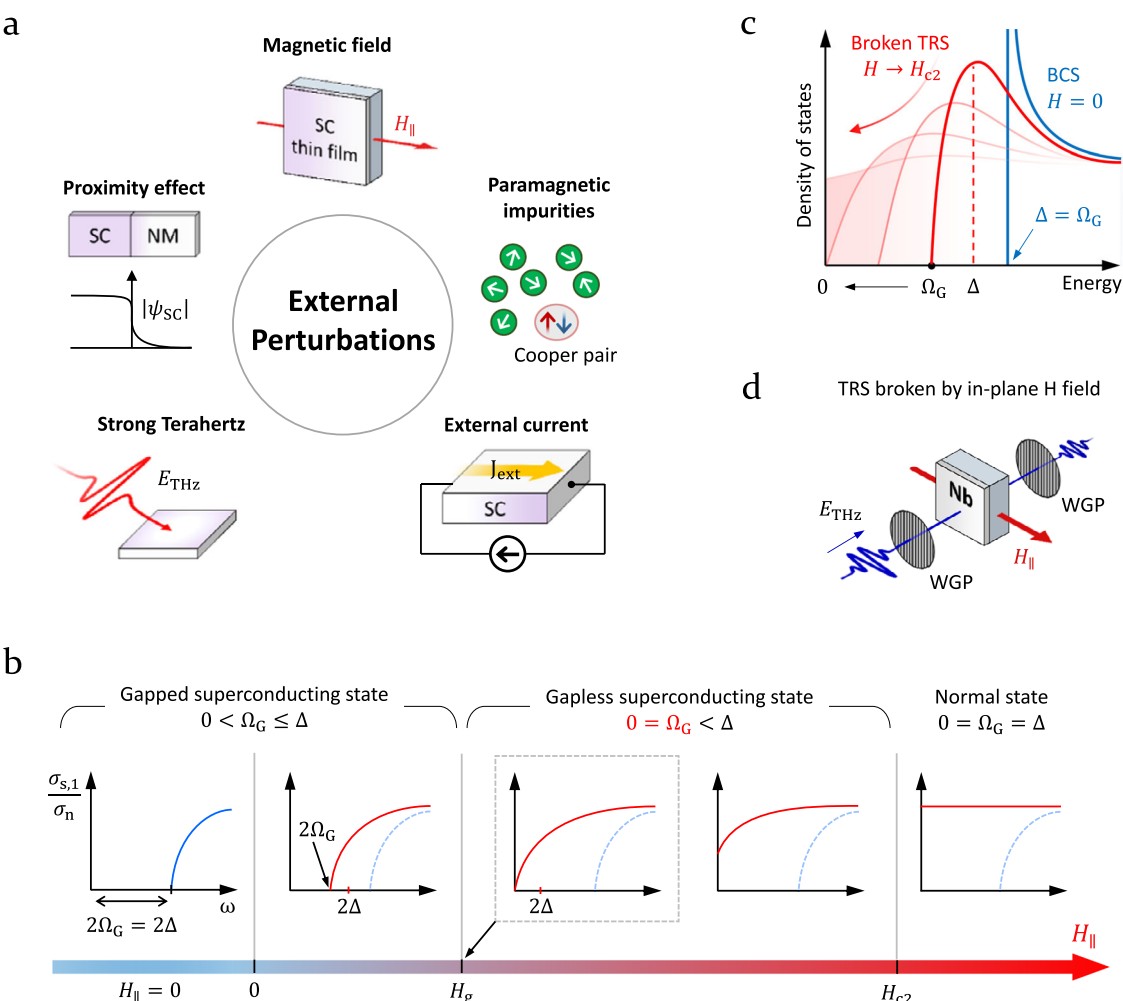

**Fig. 1 | Nonthermal pair-breaking sources in superconductors. a** External perturbations that break Cooper pairs and suppress superconductivity: an external magnetic field, paramagnetic impurities, an external supercurrent bias, ultrastrong terahertz (THz) pulses, and normal metals in close proximity. The first three of these break time reversal symmetry (TRS). SC is superconductor, and NM is normal metal. **b** Schematic diagrams that show the systematic suppression of superconductivity in the optical conductivity channel due to an external in-plane magnetic field ($H_\parallel$). $\sigma_{s,1}$ is the real part of the optical conductivity in the superconducting state, and $\sigma_n$ in the normal state (assumed to be constant real). See the main text for the definition of the variables $\Delta$ and $\Omega_G$. The blue curve (solid or broken) represents the optical conductivity in the absence of an external magnetic field. Systematic changes due to the magnetic field are represented by a series of red curves. **c** Evolution of the density of states of a superconductor under an in-plane magnetic field ($H_\parallel$). The blue curve represents the original BCS (Bardeen-Cooper-Schrieffer) theory prediction. A series of red curves represent the changes under the magnetic field. The red and black arrows indicate the increasing magnetic field and the decreasing spectroscopic gap, respectively. The broken red line indicates the position of the pair-broken order parameter under a weak magnetic field. **d** Schematics of terahertz time-domain spectroscopy (THz-TDS) in a magneto-optic measurement system. WGP is wire-grid polarizer. The red arrow indicates the direction of the external in-plane magnetic field, and the blue waveforms indicate the incident and transmitted THz pulses.

$H$, which is relevant to our investigation reported here, Maki's theory showed that $\Gamma$ is proportional to the squared field strength $H^2$[26]. In general, as $H$ increases, the spectroscopic gap $\Omega_G$ and the order parameter $\Delta$ both decrease (Fig. 1b). However, $\Omega_G$ decays with $H$ much faster than $\Delta$ does, eventually becoming zero as the superconductor enters the gapless regime where $\Delta$ is still finite. Within the gapless regime, $\Omega_G$ remains to be zero while $\Delta$ further decreases until it itself vanishes at the upper critical field $H_{c2}$, beyond which the superconductor becomes normal. Experimentally, while $\Omega_G$ can be, for example, read off from the optical conductivity spectrum, $\Delta$ is difficult to disclose in magnetically pair-broken states without detailed theoretical analysis.

Surprisingly, experimental access to the gapless superconducting state has been highly limited so far. In the early period of research in this field, tunneling spectroscopy was the tool of choice as it directly measures the density of states (Fig. 1c). For example, Millstein and Tinkham[5] and Levin[14] found that an external magnetic field caused a reduction of the spectroscopic gap $\Omega_G$ and a substantial broadening of

the otherwise singular peak in the density of states, as theoretically predicted by Strassler and Wyder[26], with some hints at the gapless superconducting state in In and Sn samples. The d.c. limit of the data of Millstein and Tinkham[5] revealed the spectroscopic gap $\Omega_G$ continuously moving to zero with increasing magnetic field without a trace of a finite gap. Old infrared studies were conducted on superconducting Pb films doped with relatively low concentrations of paramagnetic Gd or Mn impurities[18]. Magneto-infrared studies on superconducting NbN and NbTiN thin films confirmed suppression of the spectroscopic gap in the gapped superconducting state, but they came short of reaching the deep gapless regime[27,28]. Indeed, the complete evolution of the order parameter $\Delta$, the spectroscopic gap $\Omega_G$, and the pair-breaking parameter $\Gamma$ over the entire range of magnetic fields $H$ up to $H_{c2}$ has not been elucidated so far. Furthermore, a fully self-consistent, nonperturbative theory that gives complete information on $\Gamma(H)$ and $\Delta(H)$ is currently unavailable as well. Recently, the magnetic field dependent optical conductivity of NbN films was measured in the THz region[29], but the two-dimensional aspect of the

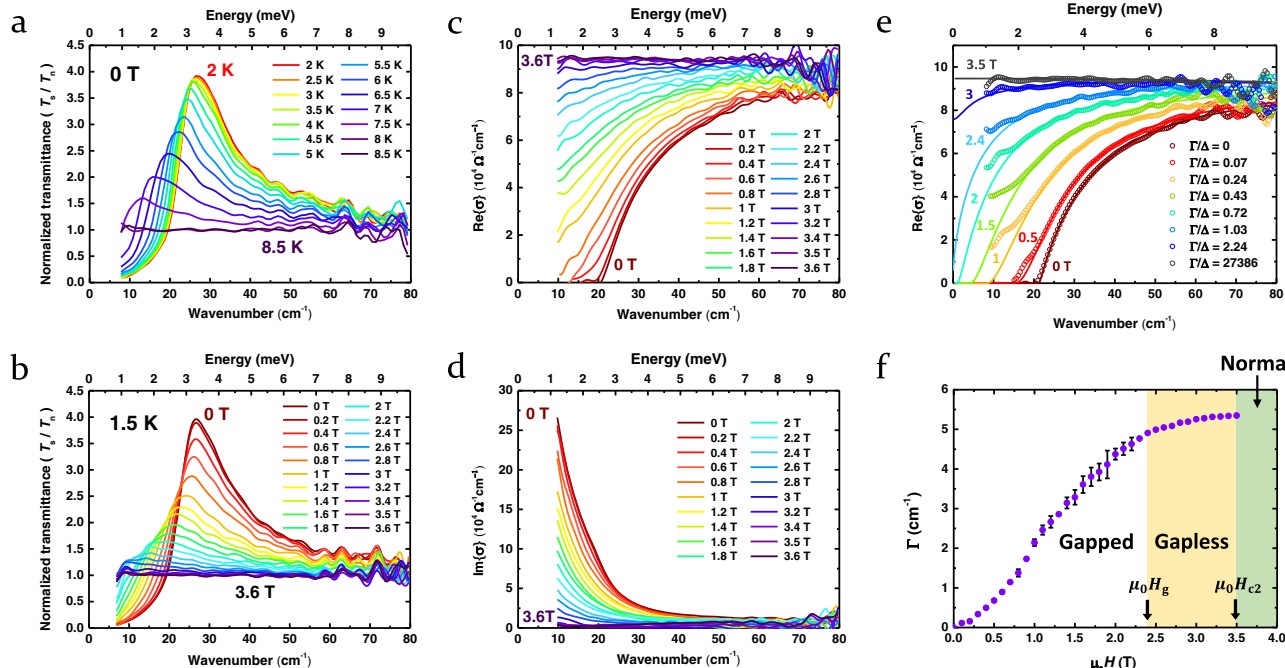

**Fig. 2 | Terahertz spectra of superconducting Nb. a** Normalized transmittance spectra of Nb at various temperatures from 2 K to 8.5 K in the absence of an external magnetic field. The transmittance $T_s$ in the superconducting state is normalized to $T_n$ in the normal state, taken at 11 K. **b** Normalized transmittance spectra of Nb under in-plane magnetic fields up to 7 T at 1.5 K. The transmittance $T_s$ in the superconducting state is normalized to $T_n$ in the normal state, taken at 7 T. **c** Real part of the optical conductivity of Nb under the in-plane magnetic field from 0 T to 3.6 T at 1.5 K. **d** Imaginary part of the optical conductivity as in **c**. **e** Real part of the optical conductivity of Nb for several magnetic fields (0–3.5 T) at 1.5 K. The pair-breaking parameter $\Gamma$ normalized to the pair-broken order parameter $\Delta(\Gamma)$ (see

text) corresponding to the magnetic field $H$ is indicated for each curve. The open circles mark experimental data, and the solid lines are fitted to the pair-breaking theory of SBW[9]. The gapped superconducting state ($0 \le \Gamma/\Delta(\Gamma) < 1$) transforms into the gapless superconducting state ($\Gamma/\Delta(\Gamma) \ge 1$) and then to the normal state ($\Gamma/\Delta(\Gamma) = 27386 \to \infty$). **f** Magnetic field dependence of the pair-breaking parameter $\Gamma(H)$ determined from the experimental optical conductivity in the gapped regime (white) and the gapless regime (yellow). The green area is in a normal state. The error bars represent the uncertainty of the SBW fit with 95% confidence intervals.

data complicated the gap closing picture. Another work by Pracht et al.[30] was perhaps the only work to date reporting the field dependent order parameter.

Here, we show that terahertz spectroscopy provides us with a rare opportunity to access and explore the gapless superconducting state of Nb thin films under an in-plane external magnetic field $H$ at low temperature. We report the detailed evolutions of the pair-breaking parameter $\Gamma$, the spectroscopic gap $\Omega_G$, the superconducting order parameter $\Delta$, and the London penetration depth $\lambda_L$ over the entire range of fields up to the in-plane upper critical field, including the gapless regime where $\Omega_G$ is zero but $\Delta$ is still finite. In particular, we present here the complete functional form of the order parameter $\Delta(H)$ for an arbitrary field $H$, for which a proper theory is yet unavailable. Our experiment shows, in the gapless region, a rapid collapse of $\Delta$ leading to an unexpected nonparabolic dependence of $\Gamma$ on $H$ and an extreme nonlinear Meissner effect.

## Results

### Terahertz spectroscopy under an in-plane magnetic field

We use terahertz time-domain spectroscopy (THz-TDS) to optically probe the evolution of the superconducting order parameter $\Delta$ and the spectroscopic gap $\Omega_G$ of superconducting Nb thin films ($T_c = 8$ K) at 1.5 K under an external in-plane magnetic field up to 7 T. The sample preparation and characterization are described in Methods. The experimental scheme of our THz-TDS measurement is depicted in Fig. 1d and further described in Methods. In this magneto-optical geometry, the in-plane upper critical field $\mu_0 H_{c2} = 3.5$ T is sufficiently low to grant easy access to the gapless regime. At the same time, the vortex excitation is suppressed as the film thickness of 58 nm in our case is comparable to the coherence length of 38 nm for Nb[31]. The

lower limit of our spectral range of nearly 1 meV (8 cm$^{-1}$) enables us to track the low-energy part of the optical conductivity $\sigma(\omega)$ in sufficient detail to the extent that the gapless features can be reliably captured and compared with theory in detail. Furthermore, our time-domain technique allows for an accurate and independent determination of the real ($\sigma_1(\omega)$) and imaginary ($\sigma_2(\omega)$) parts of $\sigma(\omega)$, without resorting to a Kramers–Kronig analysis (Methods), at each temperature ($T$) and magnetic field ($H$). From our experimental data, we extract the superconducting order parameter ($\Delta$), the spectroscopic gap ($\Omega_G$), the pair-breaking parameter ($\Gamma$), and the London penetration depth ($\lambda_L$), while acquiring the key relation between the pair-breaking parameter $\Gamma$ and the external in-plane magnetic field $H$ over the entire range of magnetic fields up to $H_{c2}$. The ensuing functional form of $\Gamma(H)$ in the deep gapless regime deviates dramatically from the $H^2$ law of Maki[6], which was derived on a perturbative pair-breaking calculation where the magnetic suppression of $\Delta$ itself was not taken into account. The gapless regime further reveals a singular divergence of the London penetration depth $\lambda_L$. Our work culminates with the complete, hitherto unknown, functional form of $\Delta(H)$ for an arbitrary magnetic field $H$, which should guide and filter future pair-breaking theories in full rigor.

We present in Fig. 2 the transmittance and optical conductivity of superconducting Nb in the terahertz region, as well as the field-dependent pair-breaking parameter. Figure 2a shows the zero-field temperature-dependent terahertz transmittance $T_s$ of Nb in the superconducting state normalized to the transmittance $T_n$ in the normal state, taken at 11 K. Starting from the flat, near-unity spectrum found at 8.5 K, slightly above $T_c = 8$ K (Supplementary Fig. 1), we observe a prominent peak developing in $T_s/T_n$ near the low-frequency end and moving toward higher frequencies with a dramatic growth in intensity as the temperature decreases. The peak position roughly

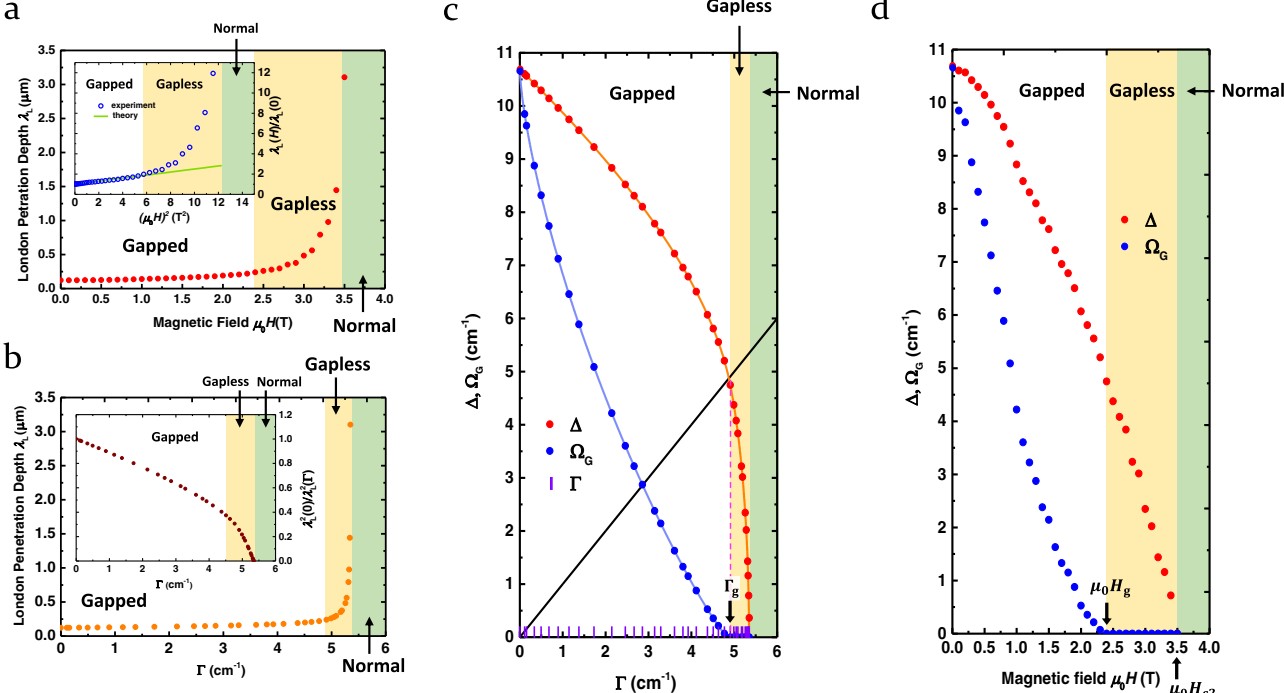

**Fig. 3 | In-plane magnetic field effect on the superconducting parameters of Nb. a** In-plane magnetic field dependence of the London penetration depth $\lambda_L(H)$ (red circles) is extracted from the experimental optical conductivity (see text) by using the sum-rule analysis (Methods). The white, yellow, and green shaded areas indicate the magnetic field ranges of the gapped, the gapless, and the normal state, respectively. The inset shows the normalized London penetration depth $\lambda_L(H)/\lambda_L(0)$ (blue circles). The green line signaling a quadratic magnetic field term is the theory of the nonlinear Meissner effect of ref. 35,36. (Methods). The deviation from this behavior in the gapless regime is an extreme nonlinear Meissner effect. **b** Dependence of the London penetration depth $\lambda_L(\Gamma)$ on the pair-breaking parameter $\Gamma$. The inset shows the normalized inverse squared penetration depth

$\lambda_L^2(0)/\lambda_L^2(\Gamma)$. This is the same as the normalized superfluid density. **c** Order parameter $\Delta$ (red circles) and spectroscopic gap $\Omega_G$ (blue circles) of Nb as functions of the pair-breaking parameter $\Gamma$. The experimentally accessed $\Gamma$ values are indicated as purple ticks on the horizontal axis. The solid curves are the SBW theory that relates $\Delta$ and $\Omega_G$ with $\Gamma$. The solid black line is the linear function with a slope of unity. The vertical dashed line (magenta) is used to locate $\Gamma_g = 4.9$ cm$^{-1}$, the pair-breaking parameter that marks the onset of the gapless regime. **d** Order parameter $\Delta$ (red circles) and spectroscopic gap $\Omega_G$ (blue circles) of Nb as functions of the applied in-plane magnetic field. The magnetic field $\mu_0 H_g = 2.4$ T that marks the onset of the gapless regime is indicated on the horizontal axis.

corresponds to $2\Delta(T)$, signaling the formation of the superconducting energy gap. At 2 K, the peak position reaches 26 cm$^{-1}$ (3.2 meV). The actual value of $2\Delta = 21$ cm$^{-1}$ (2.6 meV) at 2 K extracted from the real part of the optical conductivity (Supplementary Fig. 2) at 2 K. The temperature effect on the optical conductivity of Nb is consistent with the previous report by Pronin et al.[32] in the overlapping ranges of temperature and frequency although a higher scattering rate led to a broader lineshape at high frequencies in our case.

Figure 2b shows the normalized terahertz transmittance of Nb at 1.5 K under an in-plane external magnetic field ranging from 0 T to 3.6 T. The increasing magnetic field systematically suppresses superconductivity. The peak in $T_s/T_n$ moves to lower frequencies and weakens considerably as the magnetic field approaches the in-plane upper critical field of $\mu_0 H_{c2} = 3.5$ T. We notice that the normalized terahertz transmittance spectra flatten out, with the peak in $T_s/T_n$ essentially unidentifiable, as the magnetic field passes through 2.4 T and reaches 3.5 T. Overall, the effect of increasing magnetic field is strikingly similar to that of increasing temperature as both of them break Cooper pairs although their distinct pair-breaking mechanisms are responsible for the subtle differences in their $T_s/T_n$ spectra.

More detailed information on the magnetic pair-breaking action is available in the field-dependent optical conductivity $\sigma(\omega)$ taken at 1.5 K. In the real part $\sigma_1(\omega)$ (Fig. 2c), we see that the increasing magnetic field systematically closes the spectroscopic gap initially positioned at 21 cm$^{-1}$ (2.6 meV) while simultaneously filling in the spectral window below 80 cm$^{-1}$ (9.9 meV). As the magnetic field reaches the critical value 3.5 T, all of the missing spectral weight is recovered, and $\sigma_1$ reverts to the normal-state Drude profile, which is essentially a flat line

here as the scattering rate $\gamma = 607$ cm$^{-1}$ (75.3 meV) of Nb is much larger than twice the original gap $2\Delta = 21$ cm$^{-1}$ (2.6 meV). Likewise, in the imaginary part $\sigma_2(\omega)$ (Fig. 2d), we see that the increasing magnetic field systematically suppresses the low-frequency $1/\omega$-like component of $\sigma_2(\omega)$, directly linked to the superfluid density, that is, the strength of superconductivity. The $1/\omega$-like component also vanishes completely as the magnetic field reaches $\mu_0 H_{c2} = 3.5$ T.

## Pair-breaking parameter and its field dependence

At this point, a detailed analysis of the experimental optical conductivity spectra can be performed based on the pair-breaking theory of Skalski, Betbeder-Matibet, and Weiss (SBW)[9], an extension of the AG theory[8]. Fig. 2e shows representative experimental $\sigma_1(\omega)$ data fitted to the theoretical formula given by the SBW theory (Methods, Supplementary Fig. 3) for the $T = 0$ K case where the applied in-plane magnetic field $H$ is matched one-to-one with the pair-breaking parameter $\Gamma$. For convenience, we present in Fig. 2e the normalized value of $\Gamma/\Delta$ for a given magnetic field $H$ wherein $\Delta$ is understood to be $\Delta(T = 0, \Gamma)$ approximated by $\Delta(T = 1.5\text{K}, \Gamma)$ in the present case. We see excellent overall agreement between experiment and theory. In particular, we notice that at $\mu_0 H_g = 2.4$ T ($\Gamma_g/\Delta = 1.03 \to 1$), the spectroscopic gap closes. $H_g$ and $\Gamma_g$ represent the in-plane magnetic field and the pair-breaking parameter, respectively, which mark the onset of the gapless regime in Nb. Beyond that point, the real part $\sigma_1(\omega)$ of the optical conductivity appears to reach a finite value at zero frequency, as expected for the gapless regime. Incidentally, we note that the real part of the optical conductivity exhibits extra subgap absorption at low frequencies that seems to deviate from the SBW theory. There is an

interesting theoretical explanation of this feature that incorporates the exchange of collective excitations between the Cooper pairs broken apart by electromagnetic field[33]. This potentially important observation merits further research in the future.

Finally, at the critical pair-breaking parameter $\Gamma_c$ corresponding to $\mu_0 H_c = 3.5\,\text{T}(\Gamma_c/\Delta = 27386 \rightarrow \infty)$, Nb turns normal. Therefore, we can characterize the gapped regime ($0 \leq \Gamma < \Gamma_g$, or $0 \leq H < H_g$), the gapless regime ($\Gamma_g \leq \Gamma < \Gamma_c$, or $H_g \leq H < H_{c2}$), and the normal state ($\Gamma_c \leq \Gamma$, or $H_{c2} \leq H$) in terms of the pair-breaking parameter $\Gamma$ or the corresponding magnetic field $H$. The resulting correspondence between $\Gamma$ and $H$, established via a comparison of $\sigma_1(\omega)$ between theory and experiment, is presented in Fig. 2f. From this analysis, we can determine $\Gamma_g = 4.9\,\text{cm}^{-1}$ and $\Gamma_c = 5.35\,\text{cm}^{-1}$ corresponding to $\mu_0 H_g = 2.4\,\text{T}$ and $\mu_0 H_{c2} = 3.5\,\text{T}$, respectively. $\Gamma$ increases monotonically with the magnetic field, first starting with $H^2$-like behavior at low fields (Fig. 2f, Supplementary Fig. 4a), slows down as the gapped regime is traversed, and finally saturates quickly in the gapless regime. The initial $H^2$-like behavior is consistent with the experimental reports of Xi et al.[27] and Chauhan et al.[34] and Maki's theory[6] predicting $\Gamma = bH^2 = \tau_{tr} v_F^2 (eHd)^2 / 18\hbar c^2$ where $\tau_{tr}$ is the transport collision time, $v_F$ is the Fermi velocity, $d$ is the film thickness, $\hbar$ is the Planck constant, and $c$ is the speed of light in vacuum. However, our data go far beyond this perturbative regime, covering the entire range of magnetic fields to yield the complete functional form of $\Gamma(H)$. We stress that the whole $\Gamma$ vs. $H$ relationship has been so far unknown and that a proper theory for $\Gamma(H)$ valid for an arbitrary magnetic field $H$ has not been proposed, to the best knowledge of the authors.

**Complete field dependence of the superconducting parameters**

We are now in a position to find out the complete $\Gamma$- and $H$-dependence of the key superconducting parameters such as the London penetration depth $\lambda_L$, the spectroscopic gap $\Omega_G$, and the superconducting order parameter $\Delta$. The London penetration depth $\lambda_L$ of Nb at 1.5 K (Fig. 3a), extracted by using a sum rule analysis (Methods) and normalized to $\lambda_L(H=0)$, exhibits markedly contrasting behavior in the gapped and gapless regimes. In the gapped regime, a quadratic field dependence is noted (see Fig. 3a inset), while in the gapless regime, the deviation from this quadratic form is quite clear. Theoretically, the $H^2$ dependence was suggested by Yip and Sauls based on their perturbative calculations in the weak field approximation for a nonlinear Meissner effect in s-wave superconductors[35,36]. Our data show that this scheme breaks down in the gapless regime where the order parameter $\Delta$ itself rapidly decays with $H$. The deviation from the $H^2$ behavior represents an extreme nonlinear Meissner effect in which the superfluid density $n'_s$ itself is suppressed by an external magnetic field (Methods). With the $\Gamma(H)$ relation given in Fig. 2f, we can plot $\lambda_L$ vs. $\Gamma$ in Fig. 3b. The resulting plot embodies the experimentally sampled predictions of the SBW theory. Again, the gapped and gapless regimes exhibit markedly different behavior. The inset of Fig. 3b shows the normalized inverse squared London penetration depth $\lambda_L^2(H=0)/\lambda_L^2(H)$, which corresponds to the normalized squared superconducting plasma frequency $\omega_{ps}^2(H)/\omega_{ps}^2(H=0)$ (proportional to the normalized superfluid density). This quantity initially decreases linearly with $\Gamma$ in the gapped regime but rapidly falls in the gapless regime.

Figure 3c shows the order parameter $\Delta$ and the spectroscopic gap $\Omega_G$ for each value of $\Gamma$ determined within the SBW theory where our set of $\Gamma$ values has been acquired by fitting the experimental optical conductivity to the SBW theory. The purple vertical ticks on the horizontal axis of Fig. 3c represent the $\Gamma$ values sampled in our measurement while the red and blue circles represent the corresponding $\Delta$ and $\Omega_G$ values, respectively, lying on the theoretical curves of $\Delta(\Gamma)$ (solid red curve) and $\Omega_G(\Gamma)$ (solid blue curve), respectively. The solid black line of a slope of unity enables us to locate the intersection between $\Gamma$ and $\Delta(\Gamma)$,

which in turn locates $\Gamma_g$ where $\Gamma_g = \Delta(\Gamma_g)$ marks the onset of the gapless regime. We found $\Gamma_g = 4.9\,\text{cm}^{-1}$, as obtained above, and labeled it on the horizontal axis (Fig. 3c). Finally, the all-important $\Omega_G(H)$ and $\Delta(H)$ relations can be found by using the $\Gamma(H)$ relation already established above (Fig. 2f) and are presented in Fig. 3d. The plot shows that $\Omega_G$ decays with the magnetic field much more rapidly than $\Delta$ does, collapsing to zero at the gapped-gapless boundary. In particular, while $\Omega_G$ is dramatically suppressed as the magnetic field enters the gapless regime ($\mu_0 H_g = 2.4\,\text{T}$), $\Delta$ initially falls quadratically but then decreases more slowly than in the initial trend. The initial quadratic decrease is consistent with the perturbative result of Maki[6] for a weak field (a weak pair-breaking) that showed $\Delta - \Delta_{00} = -\pi\Gamma/4$ for small $\Gamma/\Delta$, where $\Delta_{00}$ is the order parameter at zero temperature and zero field with $\Gamma$ proportional to $H^2$. It is rather surprising that $\Delta$ crosses the gapped-gapless boundary quite smoothly without any apparent anomalies. In fact, the high-field behavior of $\Delta$ seems to be approximately modeled by another quadratic dependence (Supplementary Fig. 4b). The data presented in Fig. 3d constitute the central results of our investigation. In essence, we have experimentally captured the complete evolution of the superconducting order parameter $\Delta(H)$ and the spectroscopic gap $\Omega_G(H)$ for the entire magnetic field range that includes the BCS case ($H = 0$), the gapped regime ($0 \leq H < H_g$), the gapless regime ($H_g \leq H < H_{c2}$), and the normal state ($H \geq H_{c2}$).

## Discussion

At present, a fully self-consistent theory describing the entire evolution of the key superconducting parameters such as $\Gamma$, $\Delta$, and $\Omega_G$ with magnetic field $H$ is lacking. The SBW theory introduced the pair-breaking parameter $\Gamma$ directly related to the concentration of paramagnetic impurities, but $\Gamma$ can be flexibly associated with other pair-breaking agents such as a magnetic field. However, this $\Gamma(H)$ relation is generally only given on a perturbative basis in which the magnetic suppression of the superconducting order parameter itself is ignored. A proper theory should elucidate the key relationship $\Gamma(H)$ for arbitrary magnetic fields $H$ and solve for $\Delta$ and $\Omega_G$ self-consistently as a function of the magnetic field $H$, extending the BCS theory, strictly valid for $H = 0$ only. In this article, we have presented experimental data to test and guide such future theories. Apart from that, the novel gapless superconducting state explored in this study should be useful in understanding the extreme limit of superconductivity in connection with novel gapless states in topological materials that are under extensive scrutiny these days. The gapless superconducting state, as presented here, can be regarded as the two-dimensional (2D) nodal surface version of gapless point nodes in Dirac and Weyl semimetals and gapless nodal lines in topological nodal semimetals. One can even envision constructing heterostructures of these nodal materials and trying to identify interaction and competition among disparate, massively colliding quantum orders.

In the light of the modern perspective on topological quantum phase transitions, the gapped to gapless transition in magnetically pair-broken BCS superconductors corresponds to a Lifshitz-type of the order $2\frac{1}{2}$ as demonstrated by Yerin et al.[12]. The corresponding topological invariant characterizing the density of states manifold turns out to be the Euler-Poincare characteristic. This parameter changes from $\chi = 0$ for the gapped state to $\chi = 1$ for the gapless state. It will be exciting to check the robustness of this topological phase transition against spatial fluctuations induced by the inhomogeneity of paramagnetic dopants or strong spin fluctuations in quantum spin liquids in close proximity. Our results should further provide accurate information on the precise magnetic field dependence of the superconducting order parameter for designing and testing

quantum sensors and actuators fabricated with Josephson-junction devices in the broad field of quantum information science and technology.

## Methods

### Sample preparation

Our Nb thin films were grown on a c-plane sapphire substrate 1304 μm-thick by DC magnetron sputter deposition. Double-side polished sapphire substrates of $10 \times 10$ mm area were fixed with polyimide tape a few inches above the Nb target (99.95% pure, 2" diameter, 1/4" thick, Kurt J. Lesker Co. Ltd.). We rotated the disk where substrates were fixed to achieve high uniformity. We prepared a vacuum of a base pressure of $3.0 \times 10^{-6}$ torr and deposited films at an Ar pressure of 3.5 mtorr. Nb films were grown at a rate of about 0.58 nm/s with 296 W (760 mA × 390 V) deposition power for 1 min 40 s. We used an atomic force microscope to measure the thickness of the Nb films. After the deposition, the resistance was measured by the four-point probe technique by controlling the temperature at a rate of 0.1 K/min. The temperature at the maximum of the first derivative of the resistance was judged as the superconducting transition temperature. The Nb thin films have a thickness of 58 nm and a critical temperature of $T_c = 8$ K. The sapphire substrates exhibited negligible absorption in the terahertz (THz) region (10–100 cm$^{-1}$ or 1.24–12.4 meV).

### Terahertz time-domain spectroscopy (THz-TDS)

The temperature- and field-dependent transmission measurement was carried out on a TERA K15 spectrometer (Menlo Systems GmbH, Germany) coupled with a helium closed-cycle magneto-optical cryostat (SpectromagPT, Oxford Instruments, UK). A femtosecond laser delivered 90-fs pulses centered at a 1560 nm wavelength at a repetition rate of 100 MHz in the Menlo system. With the SpectromagPT system, we can access temperatures ranging from 1.5 to 300 K and magnetic fields ranging from 0 to 7 T. All measurements were carried out under helium exchange gas (inside the SpectromagPT sample chamber) and dry nitrogen gas (the Menlo optical paths) to remove water vapor. The raw data from the terahertz spectrometer are THz pulse waveforms of the electric field, which are subsequently converted to complex functions of frequency through fast Fourier transform (FFT). Experimentally, we actually measure the terahertz transmission of the combination of the film and the substrate. The corresponding (complex) transmission coefficient, denoted as $\tilde{t}_{f+s}$, is normalized to the (complex) transmission coefficient $\tilde{t}_s$ for a blank substrate. Then, the optical conductivity $\tilde{\sigma}(\omega) = \sigma_1(\omega) + i\sigma_2(\omega)$ is extracted from the ratio $\tilde{t}_s/\tilde{t}_{f+s}$ by using the Tinkham formula[37,38] $\tilde{\sigma} = (\tilde{n}_s + 1)(\tilde{t}_s/\tilde{t}_{f+s} - 1)/Z_0 d$ where $\tilde{n}_s$ is the complex refractive index of the substrate, $Z_0$ is the vacuum impedance, and $d$ is the film thickness.

### Transport measurements

The transport properties were measured by using a Physical Properties Measurement System (PPMS, Quantum Design Inc.). The resistance measurements were carried out by the four-probe method under an excitation of 1 mA. A horizontal rotator (P310, Quantum Design Inc.) was used to apply an external magnetic field in the in-plane direction. The sweep rates of the temperature and the magnetic field were 0.1 K/min and 20 Oe/s, respectively.

### Theory

**Skalski—Betbeder-Maribet—Weiss theory.** In the pair-breaking theory of Skalski–Betbeder-Maribet–Weiss (SBW), the pair-breaking effect can be characterized by a single pair-breaking parameter $\Gamma$, which directly controls the order parameter $\Delta$, the spectroscopic gap $\Omega_G$, and the density of states $N$. We use the SBW theory at 0 K to determine $\Gamma$ by fitting the experimental conductivity. In the gapped

superconducting state in the weak field regime ($0 \leq H < H_g$), $\Delta$ and $\Omega_G$ are determined from the two equations $\ln(\Delta/\Delta_{00}) = -\pi\Gamma/4\Delta$ and $\Omega_G = \Delta[1 - (\Gamma/\Delta)^{2/3}]^{3/2}$ where $\Delta_{00}$ is the order parameter for zero magnetic field and zero temperature. In the gapless state in the strong field regime ($H_g \leq H < H_{c2}$), $\Omega_G = 0$, and $\Delta$ is determined from the equation $\ln(\Delta/\Delta_0) = -\ln[(\Gamma/\Delta) + \{(\Gamma/\Delta)^2 - 1\}^{\frac{1}{2}}] + (\Delta/2\Gamma)[(\Gamma/\Delta)^2 - 1]^{\frac{1}{2}} - (\Gamma/2\Delta)$. We introduce an intermediate parameter $u$ as the solution of the self-consistent equation $u\Delta = q + i\Gamma u/\sqrt{u^2 - 1}$ where $q$ is the wavenumber. The optical conductivity $\tilde{\sigma}$ for a superconductor of an arbitrary purity (e.g., for the scattering rate in the normal state $\gamma = 607$ cm$^{-1}$ in our case) can be calculated from

$$\tilde{\sigma}(\omega) = \sigma_1(\omega) + i\sigma_2(\omega) = \frac{i\omega_p^2}{4\pi\omega}\left[\int_{\Omega_G + \frac{\omega}{2}}^{\infty} dq\, \text{Re}\left(\frac{1 - A_+A_- - B_+B_-}{C_- + C_+ + i2\Gamma_2}\right)\right.$$
$$+ \int_{-\Omega_G + \frac{\omega}{2}}^{\Omega_G + \frac{\omega}{2}} dq\, \text{Re}\left(\frac{1 + \tilde{A}_-A_+ + i\tilde{B}_-B_+}{i\tilde{C}_- + C_+ + i2\Gamma_2}\right)$$
$$\left. + \frac{1}{2}\left\{\int_{\Omega_G - \frac{\omega}{2}}^{-\Omega_G + \frac{\omega}{2}} dq\left(\frac{1 + A_+A_-^* + B_+B_-^*}{-C_-^* + C_+ + i2\Gamma_2} - \frac{1 - A_+A_-^* - B_+B_-^*}{-C_-^* - C_+ + i2\Gamma_2}\right)^*\right\}\right]$$

where $A_\pm = u_\pm/(u_\pm^2 - 1)^{1/2}$, $B_\pm = 1/(u_\pm^2 - 1)^{1/2}$, $C_\pm = \Delta(u_\pm^2 - 1)^{1/2}$, $\tilde{A}_- = u_-/(1 - u_-^2)^{1/2}$, $\tilde{B}_- = 1/(1 - u_-^2)^{1/2}$, and $\tilde{C}_- = \Delta(1 - u_-^2)^{1/2}$. Here, $2\Gamma_2 = \gamma$ is the scattering rate in the normal state, and $u_\pm$ corresponds to $q_\pm = q \pm \omega/2$. The density of states is extracted from the equation $N(\omega) = N_0 \text{Re}[u/(u^2 - 1)^{1/2}]$ where $N_0$ is the density of states (per one spin) at the Fermi level. The optical conductivity and the density of states obtained by using the SBW theory are shown in Supplementary Fig. 3 and Supplementary Fig. 5, respectively. We note that, although a difference in the scattering rate modifies the optical conductivity in both the normal and superconducting states, the essential features of pair breaking present in the optical conductivity spectra do not seem to exhibit a marked difference well below the spectroscopic gap, which is systematically suppressed with the magnetic field (Supplementary Fig. 3).

**London penetration depth.** We determined the London penetration depth $\lambda_L$ by utilizing the relation $\int_0^{\infty}[\sigma_{1n}(\omega) - \sigma_{1s}(\omega)]d\omega = \omega_{ps}^2/8$ where $\sigma_{1s}$ and $\sigma_{1n}$ are the real parts of the optical conductivity in the superconducting and normal states, respectively, and $\omega_{ps} = c/\lambda_L$ is the superconducting plasma frequency. It is based on the conservation of the spectral weight according to the Ferrell–Glover–Tinkham sum rule[39]. The London penetration depth can be also extracted from the imaginary part of the optical conductivity as long as the latter quantity is accurately determined without using the Kramers–Kronig relations as in our case. Therefore, we also employed the method of reading off the penetration depth from $\sigma_2$, that is, by extracting the penetration depth from the low-frequency asymptotic form (i.e., proportionality to inverse frequency) of $\sigma_2$ (Supplementary Fig. 6).

**Nonlinear Meissner effect.** At finite temperature, an thermally excited population of counter-moving quasiparticles produces a paramagnetic current, which reduces the Meissner screening. As the superfluid velocity $v_s$ increases, the formula for the supercurrent $J_s$ needs to be modified because the high-order corrections for the population difference become substantial for the order parameter $\Delta$ due to the pair-breaking effect. Yip and Sauls[35,36] have suggested a general expression for the supercurrent in an s-wave superconductor where the relevant form of the supercurrent density is $J_s = -en_s'(T)v_s[1 - \beta_1(T)(v_s/v_c)^2]$ where $-e$ is the electron charge, $n_s'$ is the superfluid density, $\beta_1$ is a temperature-dependent coefficient at low temperature, and $v_c$ is the bulk critical velocity. Yip and Sauls gave $\beta_1(T) \sim e^{-\Delta(0)/k_BT}$ and $v_c(T) = \Delta(T)/mv_F$ where $m$ is the electron mass and $v_F$ is the Fermi

velocity. Since $v_s$ is proportional to the vector potential and hence to the external magnetic field, the nonlinearity appears in the field-dependent London penetration depth. Specifically, Yip and Sauls proposed $\lambda(H,T)/\lambda(0,T) = 1 + \beta_1(T)[H/H_0(T)]^2$ where $H_0(T) = e\lambda(T)/cv_c(T)$. In the gapless regime investigated in the present work, $n_s'$ itself is reduced due to pair-breaking, which leads to an extreme nonlinear Meissner effect where the $H^2$ behavior of $\lambda(H,T)$ breaks down.

## Data availability

All data will be readily available upon request. Expert assistance and guide to the pertinent data will be provided for the best interest of the researchers who wish to use our data for their prospective research.

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

## Acknowledgements

J.H.K (Jae Hoon Kim) was supported by the Samsung Science and Technology Foundation Grant SSTF-BA2102-04 and by the National Research Foundation of Korea (Grant No. NRF-2017R1A5A1014862 (SRC Program: vdWMRC Center)). Y. J. was supported by the National Research Foundation of Korea (Grant No. NRF-2019R1A2C1089017) and BrainLink program funded by the Ministry of Science and ICT through the National Research Foundation of Korea (2022H1D3A3A01077468). K.I.S. was supported by Institute for Basic Science (IBS-R011-D1). Y.J.C.

was supported by the National Research Foundation of Korea (Grant No. NRF-2022R1A2C1006740). J.E.L. was supported by the National Research Foundation of Korea (Grant No. NRF-2022R1A6A3A01086272) and by the Yonsei University Research Fund(Post Doc. Researcher Supporting Program) of 2022 (project no.: 2022-12-0034).

## Author contributions

J.H.K., J.E.L., and K.I.S. conceived the projects. Y.J. and J.C. synthesized and characterized the samples. J.E.L. and J.H.K. conducted terahertz spectroscopy measurements and spectral analysis. K.I.S. and T.S.J contributed to the data analysis. Y.J.C. and J.H.K. (Jong Hyuk Kim) performed the transport measurements on thin films. J.E.L., J.C., K.I.S., Y.J., and J.H.K. wrote the manuscript with contributions and comments from the other authors.

## Competing interests

We declare that none of the authors have competing financial or non-financial interests as defined by Nature Portfolio.
