## [Peer Review File · Nature Communications]

Gapless superconductivity in Nb thin films probed by terahertz spectroscopyREVIEWER COMMENTS

Reviewer #1 (Remarks to the Author):

The authors study the time domain terahertz response of thin Nb films. By applying in plane magnetic fields they continuously control the pair breaking rate. Their experimental probe gives them full information about the optical conductivity in the low energy window relevant to superconductivity. They fit their optical response to the Skalski-Betbeder-Matibet (SBM) theory to obtain the effective parameters. They identify the regime of "gapless superconductivity" and find it to be quite wide and stable in terms of the magnetic field, which is somewhat surprising. They also measure the London penetration depth in this regime.

I find the paper very interesting. The study of the gapless region is extremely important because its properties are largely elusive, while there are so many superconductors that seem to be in this regime, especially in the presence of magnetic fields. Therefore, I recommend publication in Nature Communications and hope this paper will ignite further research into this phase. However, I do have a comment for the authors to consider:

The real part of the conductivity in Fig. 2.e, which is fitted to SBM theory shows clear deviation at low frequency. For example, the curve $\Gamma/\Delta = 0.43$ is deemed to be gapped by the high frequency regime, while it is not clear that it is according to the low-frequency regime. Is there some experimental artifact? Or is it possible there is a deviation from the theory? In any case, I could not find a discussion about this anywhere. I think the authors should add such a discussion where possible explanations are discussed.

Reviewer #2 (Remarks to the Author):

Report

Lee et al. measured the optical conductivity of Nb thin films in the THz frequency range as a function of temperature and magnetic field. The main focus of their work is on the analysis of the field dependent data with the Skalski-Betbeder-Matibet-Weiss (SWB) pair breaking theory which is an extension of the Abrikosov-Gor'kov (AG) theory. Lee et al. show that the spectra can be analyzed with the SWB model to extract the order parameter and the spectroscopic gap as a function of the pair breaking parameter. By investigating the evolution of the two parameters, Lee et al. identify a gapped and a gapless regimes in the superconducting state of Nb thin films.

I would like to give my recommendation for publishing this manuscript in Nature Communications, however I am concerned about several things and ask Lee et al. to consider and reply to my comments below:

1. I am very much worried that I did not see any reference to the work of Pronin et al., Phys. Rev. B 57, 14416 (1998), regarding THz spectroscopy of Nb thin films. Lee et al. should at least compare the temperature dependence and sample properties between their work and that of Pronin et al. For example the scattering rate of the films in Lee et al. work is higher by a factor of 4 than that measured by Pronin et al. It suggests that the films in Lee et al. work are dirtier and could affect the analysis of the results, as for example a possible effective medium modeling.

2. The magnetic field dependent optical conductivity spectrum of several interesting superconductors, in particular disordered superconductors, were measured in the THz frequency range already.

I refer Lee et al. to:

Šindler et al., Phys. Rev. B 105, 014506 (2022)

Pracht et al., Phys. Rev. B 86, 184503 (2012)

The work cited above deal with moderately disordered NbN and TiN thin films. Nevertheless I think that it would be relevant for the discussion of Lee et al.

Can the authors comment on these references ? Why should the Maxwell-Garnett effective medium model cannot be used here ? If it can be used, then what would be the outcome of this procedure in respect to the spectroscopic gap and the order parameter ?

3. Lee et al. explain the procedure to extract the London penetration depth from the Ferrell-Glover-Tinkham sum rule. However, I find it less relevant here since the spectrum measured by Lee et al. is not wide enough for the use of the FGT sum rule. In fact, the penetration depth can be extracted from the imaginary part of the optical conductivity since (as the authors argue correctly) it was measured directly without the need to introduce the KK relations. Can the authors explain why did they choose the FGT sum rule procedure instead of using σ_2 ?

4. Based on comment 2 above, one can consider that the magnetic field is creating areas in the film which are gapped along with areas which are gapless (considering the higher scattering rate of this sample compared to previous works) therefore invoking the question, whether effective medium theories can be used here as well? Can the authors comment on that issue ?

5. The authors cannot measure to very low frequencies but nevertheless claim that the system has no visible gap in their spectra. What if the gap is somewhere in the microwave GHz regime or at lower energies ? What if this is not a gapless regime but a very low energy gap ?

6. It appears from the actual spectra that there is some onset where the real part of the optical conductivity deviates from the fit (Fig. 2e) in a quite systematic way. The deviation shows up at 0.5 Tesla at about 20 cm^{-1} and then shifts down with frequency and increasing field, e.g. about 10 cm^{-1} at 2.4 Tesla. Can the authors comment on this deviation and suggest maybe a possible interpretation ?

7. Another possible theoretical work to consider in the view of my previous comments is the model by Fominov et al., Phys. Rev. B 84, 224517 (2011), which can explain the aforementioned sub gap absorption. I ask Lee et al. to comment on this issue.

8. I would not label the axis "real conductivity" and "imaginary conductivity" as in Fig. 2 since it seems as if one of the observables is fictitious. I would rather label it as " σ_1 " and " σ_2 " or call it "real part of the conductivity" and "imaginary part of the conductivity" or " $\text{Re}\{\sigma\}$ " and " $\text{Im}\{\sigma\}$ ".

Reviewer #1 (Remarks to the Author):

The authors study the time domain terahertz response of thin Nb films. By applying in plane magnetic fields they continuously control the pair breaking rate. Their experimental probe gives them full information about the optical conductivity in the low energy window relevant to superconductivity. They fit their optical response to the Skalski-Betbeder-Matibet (SBM) theory to obtain the effective parameters. They identify the regime of "gapless superconductivity" and find it to be quite wide and stable in terms of the magnetic field, which is somewhat surprising. They also measure the London penetration depth in this regime.

I find the paper very interesting. The study of the gapless region is extremely important because its properties are largely elusive, while there are so many superconductors that seem to be in this regime, especially in the presence of magnetic fields. Therefore, I recommend publication in Nature Communications and hope this paper will ignite further research into this phase. However, I do have a comment for the authors to consider:

We thank the reviewer for his/her positive comments and encouraging remarks.

The real part of the conductivity in Fig. 2.e, which is fitted to SBM theory shows clear deviation at low frequency. For example, the curve $\Gamma/\Delta = 0.43$ is deemed to be gapped by the high frequency regime, while it is not clear that it is according to the low-frequency regime. Is there some experimental artifact? Or is it possible there is a deviation from the theory? In any case, I could not find a discussion about this anywhere. I think the authors should add such a discussion where possible explanations are discussed.

Reply: We thank the reviewer for pointing this out and giving us an opportunity to address the issue in our revised manuscript. First of all, we check that these anomalous features were not experimental artifacts e.g. due to a low signal intensity. The signal intensity is reasonably strong, and the phase shift spectrum is stable around the frequency band containing the anomalous features. Furthermore, there is some systematics in these anomalies e.g. in terms of the "lift-off" frequency position varying with the applied magnetic field. Therefore, we believe that these are magnetic-field effects that have not been addressed in the SBW theory. Indeed, the SBW theory is based on an "averaged" propagator and hence cannot address fluctuations in the order parameter as stated in their original paper. This naturally leads to a possibility of collective excitations. In our opinion, the paper by Seibold et al. [Phys. Rev. B 96, 144507 (2017)] convincingly demonstrates that a refined random-phase-approximation (RPA) calculation can explain the observed extra "subgap" absorption by introducing the exchange of collective excitations between the Cooper pairs broken apart by e.m. field. Obviously, the applied magnetic field, which itself breaks Cooper pairs, can modulate this exchange effect. Please see the figure below copied from the paper by Seibold et al. (Fig. 1 (c)).

As the reviewer suggested, we added (p. 7) a discussion on the anomalous features in our revised manuscript in the aforementioned context and cited (Ref. 33) the paper by Seibold et al.

Reviewer #2 (Remarks to the Author):

Report

Lee et al. measured the optical conductivity of Nb thin films in the THz frequency range as a function of temperature and magnetic field. The main focus of their work is on the analysis of the field dependent data with the Skalski-Betbeder-Matibet-Weiss (SWB) pair breaking theory which is an extension of the Abrikosov-Gor'kov (AG) theory. Lee et al. show that the spectra can be analyzed with the SWB model to extract the order parameter and the spectroscopic gap as a function of the pair breaking parameter. By investigating the evolution of the two parameters, Lee et al. identify a gapped and a gapless regimes in the superconducting state of Nb thin films.

I would like to give my recommendation for publishing this manuscript in Nature Communications, however I am concerned about several things and ask Lee et al. to consider and reply to my comments below:

We thank the reviewer for his/her approval and reply to the comments below.

1. I am very much worried that I did not see any reference to the work of Pronin et al., Phys. Rev. B 57, 14416 (1998), regarding THz spectroscopy of Nb thin films. Lee et al. should at least compare the temperature dependence and sample properties between their work and that of Pronin et al. For example the scattering rate of the films in Lee et al. work is higher by a factor of 4 than that measured by Pronin et al.

We have inadvertently neglected citing the paper by Pronin et al. This paper is a seminal contribution in that they pioneered a BWO-based coherent measurement technique, demonstrating a simultaneous, independent determination of the real and imaginary parts of the optical conductivity. We cited this paper [Ref. 32] in our revised manuscript and remarked (p. 5) that there is good agreement in the overlapping frequency and temperature ranges, but that there are differences in the scattering rate and 2Δ . As to the comparison of the sample properties, we tabulate the relevant thin-film growth parameters below. Notably, the heating of the substrate during cleaning must have minimized formation of defects to the extent that the scattering rate reported in Pronin et al. is only one quarter of what we ended up with.

	Our sample	Pronin et al.
Nb Thickness	58 nm	15 nm
Substrate (Sapphire) Thickness	1.034 mm	0.45 mm
Substrate (Sapphire) Orientation	(0001) C-cut	(1102) R-cut
Cleaning Sapphire	room temperature, no heating	in situ bake at 490–500 °C
T _c	8 K	8.31 K
Scattering rate γ	607 cm ⁻¹	150 cm ⁻¹
2 Δ	21 cm ⁻¹	24 cm ⁻¹

It suggests that the films in Lee et al. work are dirtier and could affect the analysis of the results, as for example a possible effective medium modeling.

Below, we show the low-frequency zoom-in version (adopted (p. 15, Supplement p. 4) in the revised manuscript) of our old Supplementary Fig. 3, simulating (by using the SBW theory) the optical conductivity under a field-induced pair breaking, corresponding to two different cases of $\gamma=150$ cm⁻¹ and $\gamma=607$ cm⁻¹. Although a difference in the scattering rate modifies the optical conductivity in both normal and superconducting states, the essential features of pair breaking present in the optical conductivity spectra do not seem to exhibit a marked difference well below the spectroscopic gap, which is systematically suppressed with magnetic field. Based on this observation, we presume that the pair-breaking pattern found at low frequencies is essentially independent of the scattering rate on the order of a few hundred cm⁻¹.

Therefore, we believe that our analysis based on the SBW theory will not suffer critically from this small difference in the scattering rate. We also note that high scattering and dirty-limit situations are assumed in the AG and SBW theory where a strong spin-orbit scattering is assumed. As to the effective-medium issue, please see below (Reply 2).

Q2.

The magnetic field dependent optical conductivity spectrum of several interesting superconductors, in particular disordered superconductors, were measured in the THz frequency range already.

I refer Lee et al. to:

Šindler et al., Phys. Rev. B 105, 014506 (2022)

Pracht et al., Phys. Rev. B 86, 184503 (2012)

The work cited above deal with moderately disordered NbN and TiN thin films. Nevertheless I think that it would be relevant for the discussion of Lee et al.

Can the authors comment on these references?

Reply 2: Agreeing with the reviewer on the importance of these papers, we now cite them [Ref.s 29 and 30] and make (p. 4) relevant comments in the main text.

Pracht et al. is again seminal in that it is probably the first THz paper to present the energy gap vs. magnetic field over the entire field range up to the upper critical field.

Sindler et al. helps us to identify possible two-dimensional effects in 5.3 nm-thick NbN films; filling-in behavior below the energy gap even for a relatively weak magnetic field is apparent. In contrast, Xi et al. [Phys. Rev B 87, 140502(R) (2013)] did not see any signatures attributable to such two-dimensional features in their 70 nm-thick NbN films. Only a rigid shift of the energy gap is reported. Therefore, we can safely regard Sindler et al. and Xi et al. as two- and three-dimensional, respectively. Sindler et al., in a two-dimensional setting, addresses strong disorder and proximity to SIT as a source for a rather peculiar magnetic field effect, eventually invoking effective medium analysis. However, Xi et al. does not need to consider such an approach as their data are explained successfully within the AG theory.

Why should the Maxwell-Garnett effective medium model cannot be used here?

We believe that our case with 58 nm-thick Nb films is quite similar to the case of Xi et al. with 70 nm-thick NbN films. Experimentally, we do not observe the aforementioned formation of a filling-in or a “plateau” in the real part of the optical conductivity below the energy gap at a relatively weak magnetic field. Furthermore, we can successfully fit our data to the SBW theory, observing a rigid shift or a systematic “closing” of the energy gap (even for a rather strong magnetic field nearing 17 % of the upper critical field). Therefore, we believe that an effective-medium approach is not necessary.

If it can be used, then what would be the outcome of this procedure in respect to the spectroscopic gap and the order parameter?

In the future, we plan to study thinner Nb films and/or introduce irradiation-induced disorder and track the magnetic field-induced pair breaking. Sindler et al. will be a valuable reference for that work. We note that Xi et al. [Phys. Rev B 87, 184503 (2013)] in fact investigated the field dependence of the superconducting order parameter and the spectroscopic gap under an out-of-plane field where both the normal and superconducting components are present (due to vortex excitations). In that study, the superconducting component was treated with the AG theory while the normal component was treated as a lossy conductor. Then the two components were combined within the Maxwell-Garnett effective medium theory, which leads to good agreement with experiment. In this form, then, the effective medium theory and the pair-breaking theory can be combined successfully and effectively, and the meaning of the spectroscopic gap and the order parameter are well retained within the superconducting component.

3. Lee et al. explain the procedure to extract the London penetration depth from the Ferrell-Glover-Tinkham sum rule. However, I find it less relevant here since the spectrum measured by Lee et al. is not wide enough for the use of the FGT sum rule. In fact, the penetration depth can be extracted from the imaginary part of the optical conductivity since (as the authors argue correctly) it was measured directly without the need to introduce the KK relations. Can the authors explain why did they choose the FGT sum rule procedure instead of using σ_2 ?

Reply 3: We adopted the FGT sum rule method as it is one of the standard ways of extracting the London penetration depth. However, as the reviewer pointed out, the London penetration depth can also be extracted from the imaginary part of the optical conductivity as long as it is accurately determined without using Kramers-Kronig relations. In order to comply with the reviewer’s comment, we also extracted the quantity from the low-frequency asymptotic form (i.e. proportionality to inverse frequency) of the imaginary part of the optical conductivity. In the revised manuscript, we presented (p. 16, Supplement p. 7) both sets of data with some explanations.

4. Based on comment 2 above, one can consider that the magnetic field is creating areas in the film which are gapped along with areas which are gapless (considering the higher scattering rate of this sample compared to previous works) therefore invoking the question, whether effective medium theories can be used here as well? Can the authors comment on that issue?

Reply 4: For a simple metallic superconductor such as Nb, the scattering rate derives from the elastic impurity scattering which, at low temperatures, dominates the inelastic phonon scattering. Therefore, we cannot possibly think of a specific mechanism as to how an external magnetic field alone can suddenly create spatially inhomogeneous texture of gapped and gapless areas just because our scattering rate is a factor of few higher than previously reported values. In this context, we already showed (Reply 1) a minimal influence of the scattering rate on the gapless behavior well below the energy gap. Under an out-of-plane magnetic field generating vortex excitations, however, the reviewer's comment is then quite relevant as there will be gapped (locally weak magnetic field) and gapless (locally strong magnetic field) parts due to inhomogeneous magnetic fields (weak around vortices, strong away from vortices). As we apply an in-plane magnetic field with vortex intrusion severely limited, we cannot think of a potential mechanism for inhomogeneous gap texture formation induced by a homogeneous external field. Any suggestion from the reviewer is welcome if he/she can further guide us in search of possible scenarios.

5. The authors cannot measure to very low frequencies but nevertheless claim that the system has no visible gap in their spectra. What if the gap is somewhere in the microwave GHz regime or at lower energies? What if this is not a gapless regime but a very low energy gap?

Reply 5: In principle, our terahertz spectroscopic technique cannot possibly access the "very low" frequency region such as the microwaves, and therefore the detection of a "very small" gap would be beyond our capacity. However, we cite [Ref 5] and comment on (p. 4) the paper by Millstein and Tinkham [Phys. Rev. 158, 325 (1967)], where the real part of the optical conductivity as measured by tunneling conductance spectroscopy, is presented to a very low frequency or to a very low bias voltage. In the "gapless" regime, there is no trace of a gap-like feature, and we consider this as strong evidence for the absence of a "very small" gap that evades our terahertz investigations. Please see their figure copied below.

6. It appears from the actual spectra that there is some onset where the real part of the optical conductivity deviates from the fit (Fig. 2e) in a quite systematic way. The deviation shows up at 0.5 Tesla at about 20 cm^{-1} and then shifts down with frequency and increasing field, e.g. about 10 cm^{-1} at 2.4 Tesla. Can the authors comment on this deviation and suggest maybe a possible interpretation?

Reply 6: We thank the reviewer for pointing this out and giving us an opportunity to address the issue in our revised manuscript. First of all, we checked that these anomalous features were not experimental artifacts e.g. due to a low signal intensity. The signal intensity is reasonably strong, and the phase spectrum is stable around the frequency band containing the anomalous features. Furthermore, there is some systematics in these anomalies e.g. in terms of the “lift-off” frequency position varying with the applied magnetic field. Therefore, we believe that these are magnetic-field effects that have not been addressed in the SBW theory. Indeed, the SBW theory is based on an “averaged” propagator and hence cannot address fluctuations in the order parameter, as stated in their original paper. This naturally leads to a possibility of collective excitations. In our opinion, the paper by Seibold et al. [Phys. Rev. B 96, 144507 (2017)] convincingly demonstrates that a refined random-phase-approximation (RPA) calculation can explain the observed extra “subgap” absorption by invoking the exchange of collective excitations between the Cooper pairs broken apart by e.m. field. Obviously, the applied magnetic field, which itself breaks Cooper pairs, can modulate this exchange effect. Please see the figure below copied from the paper by Seibold et al. (Fig. 1 (c)).

As the reviewer suggested, we added (p. 7) a discussion on the anomalous features in our revised manuscript in the aforementioned context and cited [Ref. 33] the paper by Seibold et al.

7. Another possible theoretical work to consider in the view of my previous comments is the model by Fominov et al., Phys. Rev. B 84, 224517 (2011), which can explain the aforementioned sub gap absorption. I ask Lee et al. to comment on this issue.

Reply 7: The paper by Fominov et al. deals with the situation where magnetic impurities are doped into a superconductor in increasing concentration. This leads to a series of characteristic behaviors coming from localized states around individual magnetic impurities, which eventually leads to an impurity band. The impurity band was then presumed to induce subgap absorption. However, our Nb thin films are essentially devoid of magnetic impurities (otherwise, T_c cannot reach 8 K as magnetic impurities will quickly suppress superconductivity). Furthermore, an external magnetic field is qualitatively different from magnetic impurities at least in the context of formation of localized states. Therefore, we have reservations as to attributing the anomalous features to the impurity-band driven extra subgap absorption. Please see our Reply 6 to Q6 above where we suggested an alternative scenario derived from many-body effects. We cited this paper while discussing the issue raised in Q6 in the revised manuscript.

8. I would not label the axis “real conductivity” and “imaginary conductivity” as in Fig. 2 since it seems as if one of the observables is fictitious. I would rather label it as “ σ_1 ” and “ σ_2 ” or call it “real part of the conductivity” and “imaginary part of the conductivity” or “ $\text{Re}\{\sigma\}$ ” and “ $\text{Im}\{\sigma\}$ ”.

Reply 8: We revised the figures with new labels in the form of “ $\text{Re}\{\sigma\}$ ” and “ $\text{Im}\{\sigma\}$ ”.

REVIEWERS' COMMENTS

Reviewer #1 (Remarks to the Author):

The authors responded to my criticism. I recommend to publish the paper in Nature Communications.

Reviewer #2 (Remarks to the Author):

Lee et al. have answered all my remarks and comments. I am satisfied with the revision done to the manuscript based on the reports and thus recommend this manuscript for publication in Nature Communication.

Reviewer #1 (Remarks to the Author):

The authors responded to my criticism. I recommend to publish the paper in Nature Communications.

We thank the reviewer for the approval and also for constructive criticism and insightful comments that improved our manuscript.

Reviewer #2 (Remarks to the Author):

Lee et al. have answered all my remarks and comments. I am satisfied with the revision done to the manuscript based on the reports and thus recommend this manuscript for publication in Nature Communication.

We thank the reviewer for the approval and also for constructive criticism and insightful comments that improved our manuscript.